# A Relational Semantics for Ockham's Modalities

Davide Falessi [1,*] and Fabien Schang [2,3,*]

1 École Pratique des Hautes Études-PSL, University of Lucerne, 6002 Lucerne, Switzerland
2 Lycée Fabert, 57000 Metz, France
3 Lycée Jean Zay, 54800 Jarny, France
* Correspondence: davide.falessi@outlook.it (D.F.); schangfabien@gmail.com (F.S.);
  Tel.: +33-06-38-33-27-49 (D.F. & F.S.)

**Abstract:** This article aims at providing some extension of the modal square of opposition in the light of Ockham's account of modal operators. Moreover, we set forth some significant remarks on the de re–de dicto distinction and on the modal operator of contingency by means of a set-theoretic algebra called numbering semantics. This generalization starting from Ockham's account of modalities will allow us to take into consideration whether Ockham's account holds water or not, and in which case it should be changed.

**Keywords:** numbering semantics; contingency; de re–de dicto; logical relations; modalities; Ockham

## 1. Introduction

The article is structured as follows.

A first, more historical part will be entirely dedicated to the set-up of Ockham's account of modal propositions and their possible readings. It will be considered as the de re–de dicto distinction, providing some informal rules regarding this distinction that can be deduced from Ockham's account (Section 2, pp. 1–4, written by D. Falessi).

Then, Ockham's account of contingency and its application to the modal squares provided in Ockham's commentary on Aristotle's *De Interpretatione* is presented. This brings us to two modal hexagons that will be drawn as generalizations of those modal squares by means of the application of contingency, as Ockham defines it (Section 3, pp. 4–9, written by D. Falessi).

Finally, a formal section will be devoted to a formal semantics of Ockham's modal statements. More especially, it will consist of a set of two kinds of logical forms, whether de re or de dicto, and a corresponding second-order logic where modalities are viewed as a dyadic predicate including properties and worlds. After devising a set-theoretical semantics, according to which the meaning of formulas corresponds to their model sets or ordered truth-conditions, Ockham's statements of (non-)contingency will then be redefined by means of an external use of negation, and our algebraic translation of logical relations will result in a complex structure, i.e., a logical icosagon (Section 4, pp. 9–16, written by F. Schang, including both Appendices A and B dedicated to a logical reformulation of modal statements starting from Ockham's account).

Needless to say, the conclusion and all the sections are the result of a common work of discussion and sharing opinions and ideas.

## 2. Ockham's Account: De dicto/De re Distinction

In medieval logic, there are two possible readings of a modal proposition. A modal proposition can be taken either in sensu compositionis (compound sense) or in sensu divisionis (divided sense) (see also [1,2] for the medieval theories of modal logic). For a fully-fledged explanation of Ockham's account of modalities, see [3,4]. We shall consider here just the de re and de dicto readings, the status of contingency as a modal operator, and the modal squares. Ockham defines the compound sense as follows:

> In the sense of composition it is always asserted that such a mode is truly predicated of the proposition corresponding to the dictum in question. For example, by means of "That every man is an animal is necessary" it is asserted that the mode "necessary" is truly predicated of the proposition "Every man is an animal", the dictum of which is "That every man is an animal" ([5] II, 9, 13–17, transl. p. 109).

Therefore, in the compound sense, what is taken under consideration is the dictum, namely the categorical proposition to which the modal operator is linked. The modal proposition is true if the dictum satisfies the requirements of the modal operator attached to it. For example, assume that there is a proposition de necessario, i.e., a proposition having necessity as a modal operator, and it is taken in the compound sense, then that proposition is true if and only if the categorical proposition or dictum is a necessary proposition, such as in the case of "every man is an animal". Hence it is called "compound sense" because the proposition is taken as a composite, as a unitary entity that can be necessary, possible, etc., based on its dictum.

Regarding the divided sense, Ockham states:

> However, the sense of division of such a proposition is always equipollent to a proposition taken with a mode and without such a dictum. For example, "That every man is an animal is necessary"; in the sense of division is equipollent to "Every man is of necessity (or necessarily) an animal". ([5] II, 9, 19–23, transl. p. 109)

The divided sense does not have a dictum, but the modal operator "divides" the subject from the predicate by introducing a changing of the copula, *within* the modal proposition. Ockham clearly maintains that a proposition in divided sense is true or false according to the references of the terms involved in the proposition. In order to evaluate if the proposition is true, it is required to go through the individuals that are denoted by the subject and to rewrite the proposition at stake in the correspondent singular propositions:

> "It is necessary that every man is white" is true in the divided sense
>
> if and only if
>
> "This man (hoc) is white" and "That man is white", etc., are all true and necessary
>
> (See [5] II, 10, 11–24).

Therefore, if it is the case that all the singular propositions are true and necessary, the proposition taken in the divided sense is true and necessary as well.

Ockham also states that the compound sense requires that the proposition is taken *materially* (*materialiter*) while it is taken *significatively* (*significative*) in the divided sense (see [5] III-1, 20, 30–38). Regarding the former, the term "materially" can be better understood looking at the well-known medieval supposition theory. A term has a material supposition (suppositio materialis) when it stands for itself, e.g., "dog has three letters". Hence, in case of the compound sense, the modal proposition "stands for itself", or better for its dictum: a proposition taken in the compound sense is true or false according to its dictum, just like "dog" is said to have three letters for that term stands for itself, i.e., for the word "dog". On the opposite side, a proposition in the divided sense is taken "significatively". This means that the truth-value of a modal proposition taken in divided sense is based on the meanings of the terms involved in that proposition. By means of the singular propositions, it can be verified whether the relation between the subject and the predicate is one that is required by the modal operator inside the modal proposition.

Finally, it can be said that the compound/divided sense distinction is based on the fact that the truth-value of a proposition changes if we consider either the subject in its relation with the predicate or the proposition as a whole. In other words, the distinction is based on a mereological distinction: what applies to the proposition as a whole does not apply to the parts of the whole, i.e., subject and predicate, taken separately and vice versa. Indeed, if a part of the proposition, say, the subject $S$, can be said to be, for instance, necessarily $P$, it does not entail that the whole proposition can be said to be necessary as well.

Hence it is quite important to distinguish between these two senses, for the same modal proposition can have different truth-values according to these different readings. For example, the proposition "it is necessary that every truth is true" is true in the compound sense, but false in the divided sense, for not every singular proposition of that proposition is necessary. There are some truths that become "stale", as Hegel would say, and are not necessary, at least in the sense of always being true. It is also possible that a modal proposition is false in the compound sense but true in the divided sense. Ockham gives this example:

> An example: "both parts of a contradiction can be true" is false in the sense of composition and true in the sense of division, since each singular is true ([6], II, q. 5, 131 67–69, transl. p. 112).

A singular proposition correspondent to a part of a contradiction is possibly true; the same for the other part of the contradiction. However, if we consider both the parts together in the propositions "both parts of a contradiction can be true", the proposition is clearly false. In other words, the fact that something is possible for the singular references of a proposition does not entail that this is also possible for the proposition in itself.

In terms of truth-conditions, it must be noticed that the truth-conditions of a proposition taken in the divided sense are actually based on the truth-conditions of the compound sense. When it is required to establish whether a proposition is true in the divided sense, it is also required to reduce that proposition to its singular propositions. Nevertheless, all the singular propositions can be only taken in the compound sense: in a singular proposition, there is nothing else that can be "divide"; that is, it is not possible to go further in the analysis of the reference, but the singular proposition is what makes the reference of a term clear. In other words, a singular proposition can be taken only in a compound sense, for it is basically an atomic proposition.

There is a also a syntactic distinction that requires some attention, especially in its relation with these two possible readings, as well as with the semantic level.

Let us take one modal operator, such as possibility. This is a cum dicto form:

$$\Diamond(\text{for } S \text{ to be } P)$$

In this form, the modal operator is attached to the dictum. The modal operator is always external, and it is always in the form "it is possible that". The dictum has this form: $S$ is in the accusative case, and $P$ is a verb in the infinitive form.

There is also a sine dicto form:

$$S \text{ is } \Diamond P$$

In this case, the modal operator is either a verb ($S$ can be $P$) or an adverb ($S$ is possibly $P$). In this form, there is not any dictum at all, but the modal operator is internal.

Now, a proposition sine dicto is always taken in the divided sense because an internal modal operator always "divides" the subject from the predicate so that the relationship between them must be always verified by looking at the singular propositions. A proposition cum dicto can exist in either the divided or compound sense. In the case of a cum dicto form, it is necessary to clarify which reading of the modal proposition is considered. As a result of that, a cum dicto proposition that is taken in the divided sense is equivalent to the same proposition sine dicto ([5], II, 10, 2–4). All in all, when there is a cum dicto form, there is an ambiguity between the divided and the compound sense. Ockham usually avoids that ambiguity by unifing the semantical level (compound and divided sense) and the syntactical level (cum dicto–sine dicto). The cum dicto form is usually taken in the compound sense, whereas the sine dicto form is taken in the divided sense. We can summarize this by using the distinction de dicto/de re as follows:

$$\underbrace{compound \ sense \ + cum \ dicto}_{\text{de dicto}} \qquad \underbrace{divided \ sense \ + sine \ dicto}_{\text{de re}}$$

Therefore, when one considers a modal proposition de dicto, this means that the modal proposition is cum dicto and taken in the compound sense. When one considers a modal proposition de re, that proposition is sine dicto and taken in the divided sense. Hereafter, we shall refer just to the modern distinction de dicto/de re to denote both the semantical aspect and the syntactical aspect included in Ockham's account.

Before considering the modal squares and their extensions, it is necessary to set up some rules aimed at describing in which case a proposition is either de re or de dicto. These rules could be called a posteriori rules because they are deduced from Ockham's account. (It is required to justify those rules from a formal point of view: this will be done in Section 4.2.3). We shall take only the case of necessity and possibility into account because Ockham's squares, which will be considered here, involve just necessity and possibility. (Note that, hereby, "possibility" is taken in its one-sided reading, i.e., possibility that is contradictory to impossibility but subaltern to necessity; see [7]). Finally, it is important to note that the set-up of these rules is based on the truth-values of the dictum, so they are basically *content-based* rules: according to the content of the dictum, Ockham expresses the correspondent modal proposition either de re or de dicto.

There are three possible truth-values of a dictum:

(a)   A dictum is true and cannot be false: ($D^\top$), e.g., "every man is an animal".
(b)   A dictum is false and cannot be true: ($D^\perp$), e.g., "every white thing is a black thing".
(c)   A dictum can be either true or false: ($D^{\top/\perp}$), e.g., "every man is white".

Given that, it can be said that:

$$\Box/\Diamond\forall/\exists(D^\top) \text{ is always de dicto}$$

In Ockham's account, propositions such as "it is necessary/possible that every/some man is an animal" are always de dicto, even if they can also be true de re. Indeed, a proposition de dicto is necessary/possible if and only if the dictum is necessary/possible. Therefore, if the dictum is true and cannot be false, it can be said to be necessary and therefore possible (ab necesse ad posse). As a result of that, when there are dicta such as "Socrates is Socrates" or "Socrates is a man", if a modal proposition is formed starting from those dicta, that will always be de dicto.

By contrast, when a dictum is false and cannot be true, we have that:

$$\Box/\Diamond\forall/\exists(D^\perp) \text{ is always de re}$$

A proposition like "every living being is a corpse" is true de re both in case of possibility and necessity, but it is not true de dicto. It is false that "it is necessary/possible that every living being is a corpse", for it is false that "it is necessary/possible that every living being is not a living being". However, it is true that "every living being can be a corpse". It is also true that "every living being is necessarily a corpse" in the sense that every living being must sooner or later die. Therefore, a modal proposition formed from an "opposition", such as "every white thing is black", is always de re.

Finally, if the dictum is true but can be false and vice versa, we have:

$$\Box/\Diamond\forall/\exists(D^{\top/\perp}) \text{ is either de dicto or de re}$$

For example, "every animal is a man" is false but can be true both de dicto and de re: "it is possible that every animal is a man" and "every animal can be a man".

## 3. Extension of Ockham's Modal Squares

Whereas the standard version of the modal square involves all four alethic modal operators at the same time, although it includes neither universals nor particulars, the modal logic of the 14th century tried to include quantified propositions within the modal square. The necessity to introduce quantified propositions and to involve all the alethic modalities produce a splitting of the standard modal square into different squares having, for example, just two modal operators in the top and bottom sides, as universals and particulars.

From his perspective, Ockham provides three squares in his commentary on *De Interpretatione*. We shall focus on two of them involving necessity and possibility.

In the first square, there is necessity on the up side (universals) and possibility on the bottom side (particulars). Assuming that the relations embodied by the traditional square do not disappear with the modern formalization, if we have that

*A* is $\forall x \, (Sx \to \Box \, Px)$;
*E* is $\forall x \, (Sx \to \Box \neg \, Px)$;
*I* is $\exists x \, (Sx \land \Diamond \, Px)$;
*O* is $\exists x \, (Sx \land \Diamond \neg \, Px)$;

the first modal square provided by Ockham is Figure 1:

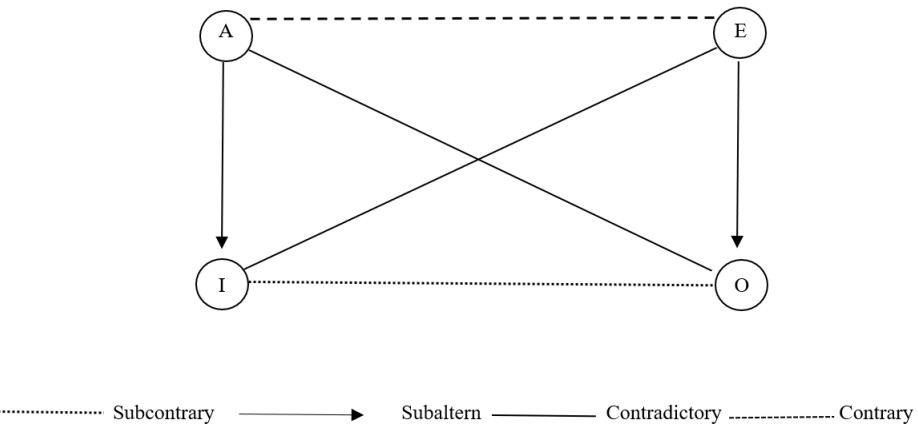

Figure 1. The first modal square provided by Ockham ([8], II, c. 7, §9, 489, 133–134).

They are all de re, but Ockham clearly says that this square can be also re-written with all the propositions in a de dicto form ("It must be known that this square is valid either if all the propositions involved are taken in a compound sense or something equivalent to a compound sense or in a divided sense or something equivalent to a divided sense." ([8], II, c. 7, §9, 489, 135–137)). Indeed, as we have said, the dictum "every/some man is white" can be both true and false ($D^{\top/\bot}$).

The same holds for the second square in which the same dictum is involved (e.g., "every/some man is white"). However, this square has possibility on the up side (universals) and necessity on the bottom side (particulars); therefore, assuming that

*A* is $\forall x \, (Sx \to \Diamond \, Px)$;
*E* is $\forall x \, (Sx \to \Diamond \neg \, Px)$;
*I* is $\exists x \, (Sx \land \Box \, Px)$;
*O* is $\exists x \, (Sx \land \Box \neg \, Px)$;

the second square is Figure 2:

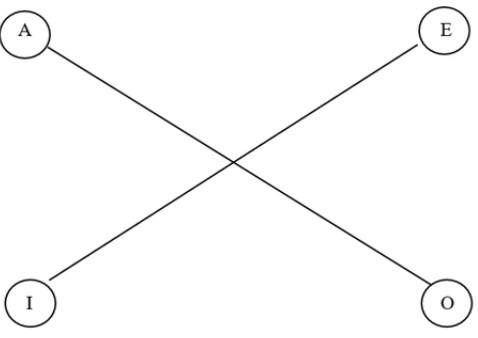

Figure 2. The second square ([8], II, c. 7, §9, 489, 133–134).

This square is a "degenerate square" which is opposed to the "classical square" (Figure 1). Indeed, Figure 2 is incomplete because the universals ("every man can be white"—"every man can not be white") are not contrary, for they can both be true the same time. In the same way, the two particulars ("some man is necessarily white"—"some man is necessarily not white") are not subcontraries because they can both be false at the same time (see [8], II, c. 7, §9, 491, 172–177).

These two squares can be extended by introducing the modal operator of contingency. First of all, it is necessary to briefly define what is contingent as a modal operator according to Ockham. (For a more abstract approach to the concepts of contingency and non-contingency, see e.g., [9].) Note that we shall consider only propositions de contingenti taken de dicto, e.g., "it is contingent that every *S* is *P*", because Ockham's account of propositions de contingenti is mainly based on just this reading of that modal proposition.

Given that $\nabla$ means "it is contingent that" and that $\Delta$ means "it is not contingent that" or "it is determinate that", Ockham defines propositions de contingenti, or having contingency as the modal operator, as follows ([5], III-3, 15, 647, 9–11; [8], II, c. 7, §9, 491, 178–492, 196):

Universal de contingenti (it is contingent that every *S* is *P*):

$$\nabla\forall x(Sx \to Px) = \Diamond\forall x(Sx \to Px) \wedge \Diamond\forall x(Sx \to \neg Px)$$
$$\nabla\forall x(Sx \to Px) = \forall x(Sx \to \Diamond Px) \wedge \forall x(Sx \to \Diamond\neg Px)$$

Particular de contingenti (it is contingent that some *S* is *P*):

$$\nabla\exists x(Sx \wedge Px) = \Diamond\exists x(Sx \wedge Px) \wedge \Diamond\exists x(Sx \wedge \neg Px)$$
$$\nabla\exists x(Sx \wedge Px) = \exists x(Sx \wedge \Diamond Px) \wedge \exists x(Sx \wedge \Diamond\neg Px).$$

First of all, note that Ockham seems to consider that, for example, the universal de contingenti de dicto is equivalent to both $\Diamond\forall x(Sx \to Px) \wedge \Diamond\forall x(Sx \to \neg Px)$ and $\forall x(Sx \to \Diamond Px) \wedge \forall x(Sx \to \Diamond\neg Px)$. This is not valid, as we shall see in Section 4.2.3. By contrast, $\Box\exists x(Sx \wedge Px) \vee \Box\exists x(Sx \wedge \neg Px) = \exists x(Sx \wedge \Box Px) \vee \exists x(Sx \wedge \Box\neg Px)$ and $\Box\forall x(Sx \to Px) \vee \Box\forall x(Sx \to \neg Px) = \forall x(Sx \to \Box Px) \vee \forall x(Sx \to \Box\neg Px)$, in the cases $\Delta\forall x(Sx \to Px)$ and $\Delta\exists x(Sx \wedge Px)$, are valid (see again Section 4.2.3). This is because of the above-mentioned content-based "rules" regarding the relationship between dicta and the modal propositions taken de dicto or de re).

Moreover, these definitions are based on one side of De Morgan's rules that Ockham knows and clearly defines in [5] ("It should also be noted that the contradictory opposite of a conjunctive proposition is a disjunctive proposition composed of the contradictories of the parts of the conjunctive (opposita contradictorie copulativae est una disiunctiva composita ex contradictoriis partium copulativae)." [5], II, 32, 348, 22–23, transl. p. 187):

$$\neg(A \wedge B) \leftrightarrow (\neg A \vee \neg B)$$

Therefore, these are the negation of the propositions de contingenti:

$$\Delta\forall x(Sx \to Px) = \Box\exists x(Sx \wedge Px) \vee \Box\exists x(Sx \wedge \neg Px)$$
$$\Delta\forall x(Sx \to Px) = \exists x(Sx \wedge \Box Px) \vee \exists x(Sx \wedge \Box\neg Px)$$
$$\Delta\exists x(Sx \wedge Px) = \Box\forall x(Sx \to Px) \vee \Box\forall x(Sx \to \neg Px)$$
$$\Delta\exists x(Sx \wedge Px) = \forall x(Sx \to \Box Px) \vee \forall x(Sx \to \Box\neg Px)$$

Ockham gives the following examples of propositions de contingenti ([5], III-3, 15, 647, 24–33; [8], II, c. 7, §9, 491, 184–492, 196):

- It is contingent that every man is an animal = it is possible that every man is an animal, and it is possible that every man is not an animal.
- It is not contingent that every man is an animal = it is necessary that some man is an animal or it is necessary that some man is not an animal. (Note that the dicta "every man is an animal" and "some man is an animal" are of the form $(D^\top)$ and hence they are de dicto).
- It is contingent that some man is white = some man can be white and some man can not be white.

- It is not contingent that some man is white = every man is necessarily white or every man is necessarily not white.

Contingency is the conjunction of opposite simultaneous possibilities. Which kind of opposition is at stake here? That one between the propositions de possibili "some man can be white" and "some man can not be white", that is to say, subcontrariety. Note indeed that two subcontraries can be conjoined, for they can be both true at the same: when it is possible to conjoin two subcontraries de possibili, there is a proposition de contingenti, "it is contingent that some man is white". Similarly, the negation of a proposition such as "it is contingent that some man is white" is equivalent to the disjunction of the negations of "some man can be white" and "some man can not be white". Therefore, we have: "every man is necessarily white" or "every man is necessarily not white". These propositions are contraries, so *either* one *or* the other can be true. This is a disjunction. In addition, Ockham states that a proposition de contingenti implies not only one proposition de possibili, but two propositions, the subcontraries de possibili ([5], III-3, 12, 640, 38–44), just as a logical conjunction $A \land B$ implies both conjuncts $A,B$. This can be summarized as follows:
De dicto

- $\nabla \forall x (Sx \to Px) \to \Diamond \forall x (Sx \to Px)$
- $\nabla \forall x (Sx \to Px) \to \Diamond \forall x (Sx \to \neg Px)$
- $\nabla \exists x (Sx \land Px) \to \Diamond \exists x (Sx \land Px)$
- $\nabla \exists x (Sx \land Px) \to \Diamond \exists x (Sx \land \neg Px)$

De re

- $\nabla \forall x (Sx \to Px) \to \forall x (Sx \to \Diamond Px)$
- $\nabla \forall x (Sx \to Px) \to \forall x (Sx \to \Diamond \neg Px)$
- $\nabla \exists x (Sx \land Px) \to \exists x (Sx \land \Diamond Px)$
- $\nabla \exists x (Sx \land Px) \to \exists x (Sx \land \Diamond \neg Px)$

By contrast, a proposition de necessario implies the negation of the proposition de contingenti ([5], III-3, 12, 639, 15–18), just like both disjuncts $A,B$ imply the logical disjunction $A \lor B$. We can therefore provide the following formalization (note that we shall provide a straightforward formalization of all the modal statements in first-order logic in Section 4):
De dicto

- $\Box \forall x (Sx \to Px) \to \Delta \forall x (Sx \to Px)$
- $\Box \forall x (Sx \to \neg Px) \to \Delta \forall x (Sx \to Px)$
- $\Box \exists x (Sx \land Px) \to \Delta \exists x (Sx \land Px)$
- $\Box \exists x (Sx \land \neg Px) \to \Delta \exists x (Sx \land Px)$

De re

- $\exists x (Sx \land \Box Px) \to \Delta \forall x (Sx \to Px)$
- $\exists x (Sx \land \Box \neg Px) \to \Delta \forall x (Sx \to Px)$
- $\forall x (Sx \to \Box Px) \to \Delta \exists x (Sx \land Px)$
- $\forall x (Sx \to \Box \neg Px) \to \Delta \exists x (Sx \land Px)$

Finally, what is contingent is said to be incompatible with (repugnans) both necessary and impossible. Therefore, contingent is what is usually called "two-sided possibility": a proposition de contingenti such as "it is contingent that some man is white" is incompatible with both "every man is necessarily white" and "every man is necessarily not white" ([5], III-3, 13, 643, 25–28).

To sum up, by introducing contingency to the squares presented above, assuming that

$A$ is $\forall x (Sx \to \Box Px)$;
$E$ is $\forall x (Sx \to \Box \neg Px)$;
$I$ is $\exists x (Sx \land \Diamond Px)$;
$O$ is $\exists x (Sx \land \Diamond \neg Px)$;
$Y$ is $\nabla \exists x (Sx \land Px)$;
$U$ is $\Delta \exists x (Sx \land Px)$;

the first figure is the following (Figure 3):

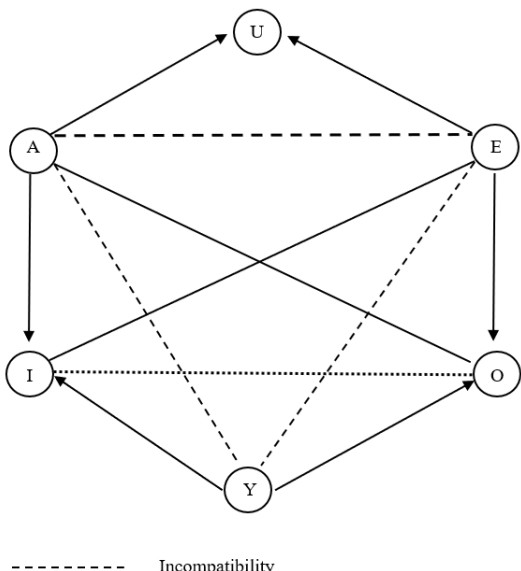

**Figure 3.** The first figure (Ockham's modal hexagon).

Moreover, assuming that:

*A* is $\forall x(Sx \rightarrow \Diamond Px)$;
*E* is $\forall x(Sx \rightarrow \Diamond \neg Px)$;
*I* is $\exists x(Sx \wedge \Box Px)$;
*O* is $\exists x(Sx \wedge \Box \neg Px)$;
*Z* is $\nabla \forall x(Sx \rightarrow Px)$;
*K* is $\Delta \forall x(Sx \rightarrow Px)$;

the second figure is the following (Figure 4):

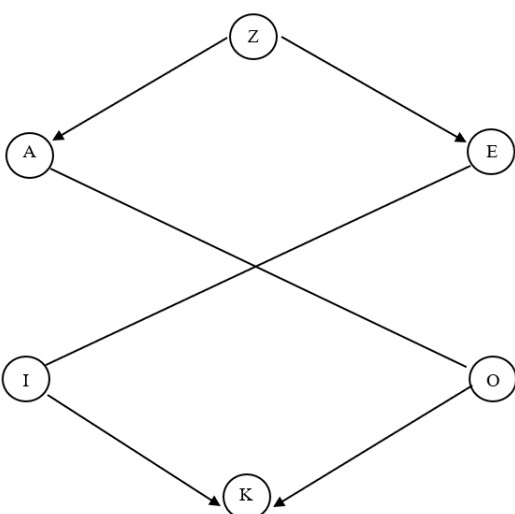

**Figure 4.** The second figure ( an incomplete version of the standard modal hexagon).

These figures are the extensions of Ockham's modal squares following Ockham's claims about contingency. Figure 3 is the standard modal hexagon provided by [10]. However, Ockham's modal hexagon (Figure 3) does not have a relation of subcontrariety among *U*, *I*, and *O* that should be present in a standard modal hexagon. Therefore, Ockham's modal hexagon is partially incomplete. Figure 4 is also an incomplete version

of the standard modal hexagon because Figure 2 is an incomplete version of the square of opposition. However, this latter figure shows that the modal hexagon in quantified modal logic is split in two different version, Figures 3 and 4. In other words, the standard modal hexagon must be redefined in order to account for all the propositions de contingenti in quantified modal logic so that we cannot admit only one version of it, but two different versions, i.e., Figures 3 and 4.

We have two final remarks. The first regards propositions de contingenti taken de re. Ockham briefly outlines an account of those propositions in his *Summa Logicae* ([5], III-3, 16, 648, 34–36). However, see also Ockham's account of modal consequentiae and syllogism, where he often uses propositions de contingenti taken de re. A proposition such as "no *S* is contingently *P*" is not equivalent to "every *S* is necessarily *P* or every *S* is necessarily not *P*", but it is equivalent to "every *S* is necessarily *P* or necessarily not *P*", so a proposition with a disjunction of the predicate *P* (propositio de disiuncto praedicato ([5], III-3, 16, 648, 35)). Note that the example provided by Ockham shows that he is aware of the fact that one cannot distribute universal quantifiers over a disjunction, as was explained clearly by [11]. Indeed, being de re, it is taken in a divided sense; hence, *S* is "separated" from *P* by the modal operator. In this way, it is *S* that is said to be contingently *P* (de re), not the whole dictum (de dicto): the subject is one, and the inherence of the predicate is affected by the modal operator because of Ockham's content-based account of de dicto and de re readings. However, Ockham does not dedicate further considerations to propositions de contingenti taken de re, and his explanation of the semantics of a proposition de contingenti is always de dicto.

Secondly, we must consider Ockham's application of the negation within the modal propositions that are either conjoined in a proposition de contingenti or disjuncted in its negation. There are two possible applications:

- Internal: e.g., "every *S* is possibly not *P*".
- External: e.g., "not every *S* is possibly *P*" = "some *S* is possibly not *P*".

Ockham takes only internal negation into account, without considering the external one. This is what makes possible our application of those propositions de contingenti within the modal squares. Indeed, let us consider the case in which "it is contingent that every *S* is *P*" would be equivalent to "it is possible that every *S* is *P*, and it is possible that some *S* is not *P*", therefore having an external negation as a second conjunct. Starting from that, the proposition "it is not contingent that every *S* is *P*" is equivalent to "it is necessary that some *S* is not *P*, or it is necessary that every *S* is *P*". As a result, contingency will not be related to all four angles of the square (A, E, O, I) but just to A and O de possibili and de necessario. Therefore, external negation seems to make it impossible to draw a standard modal hexagon in quantified modal logic, whereas internal negation helps to construct a regular hexagon of quantified modal logic. In the next sections, we shall see how, starting from Ockham's account and external negation, this would lead to an even more complicated resulting structure.

## 4. Logical Analysis

The point in the following is to set forth a systematic theory of modal statements by streamlining them into a basic logical form and then devising a corresponding formal semantics where modalities are turned into relations holding between propositions and possible worlds.

### 4.1. Syntax

Let us attempt to streamline Ockham's logical theory of quantified modal statements into the language of first-order logic in order to obtain a comprehensive formalization of it. For this purpose, these statements include two kinds of quantifiers (universal and existential) and four modal operators (necessity, possibility, non-contingency or determinacy, and contingency) that can be switched to each other and result in either de dicto or de re formulas.

A first way to account for these modal statements is ordinary language. Let $\mathcal{Q}$ = {A, E, I, O} be a set of 4 quantifying expressions ("Every is (not) …", "Some is (not) …"), and let $\mathcal{M} = \{\Box, \Diamond, \Delta, \nabla\}$ be a set of 4 modal operators (necessity, possibility, non-contingency or determinacy, and contingency). This results in a cardinal set of $4 \times 4 \times 2 = 32$ modal statements; in the first group (i)–(xvi) of statements, the quantifying expressions occur de dicto (with a broad scope), whereas they occur de re (with a narrow scope) in the second group (xvii)–(xxxii).

<div align="center">

De dicto statements
$\mathcal{MQ}$: "It is … that … S is … P"

</div>

$\Box\mathcal{Q}$: "It is necessary that … S is … P"
    (i)    $\Box$A: "It is necessary that every S is P"
    (ii)   $\Box$E: "It is necessary that every S is not P"
    (iii)  $\Box$I: "It is necessary that some S is P"
    (iv)  $\Box$O: "It is necessary that some S is not P"

$\Diamond\mathcal{Q}$: "It is possible that … S is … P"
    (v)    $\Diamond$A: "It is possible that every S is P"
    (vi)   $\Diamond$E: "It is possible that every S is not P"
    (vii)  $\Diamond$I: "It is possible that some S is P"
    (viii) $\Diamond$O: "It is possible that some S is not P"

$\Delta\mathcal{Q}$: "It is determinate that … S is … P"
    (ix)    $\Delta$A: "It is determinate that every S is P"
    (x)    $\Delta$E: "It is determinate that every S is not P"
    (xi)   $\Delta$I: "It is determinate that some S is P"
    (xii)  $\Delta$O: "It is determinate that some S is not P"

$\nabla\mathcal{Q}$: "It is contingent that … S is … P"
    (xiii) $\nabla$A: "It is contingent that every S is P"
    (xiv) $\nabla$E: "It is contingent that every S is not P"
    (xv)  $\nabla$I: "It is contingent that some S is P"
    (xvi) $\nabla$O: "It is contingent that some S is not P"

<div align="center">

De re statements
$\mathcal{QM}$: "… S is … … P"

</div>

A$\mathcal{M}$: "Every S is … P"
    (xvii)  A$\Box$: "Every S is necessarily P"
    (xviii) A$\Diamond$: "Every S is possibly P"
    (xix)  A$\Delta$: "Every S is determinately P"
    (xx)   A$\nabla$: "Every S is contingently P"

E$\mathcal{M}$: "Every S is … not P"
    (xxi)  E$\Box$: "Every S is necessarily not P"
    (xxii) E$\Diamond$: "Every S is possibly not P"
    (xxiii) E$\Delta$: "Every S is determinately not P"
    (xxiv) E$\nabla$: "Every S is contingently not P"

I$\mathcal{M}$: "Some S is … P"
    (xxv)   I$\Box$: "Some S is necessarily P"
    (xxvi)  I$\Diamond$: "Some S is possibly P"
    (xxvii)  I$\Delta$: "Some S is determinately P"
    (xxviii) I$\nabla$: "Some S is contingently P"

O$\mathcal{M}$: "Some S is … not P"
    (xxix)  O$\Box$: "Some S is necessarily not P"
    (xxx)   O$\Diamond$: "Some S is possibly not P"
    (xxxi)  O$\Delta$: "Some S is determinately not P"

(xxxii)   O∇: "Some *S* is contingently not *P*"

A second way to account for the meaning of modal statements is by means of first-order logic, in order to show subsequently which are logically equivalent. Formal logic renders the previous informal statements (i)–(xxxii) in terms of quantified expressions and their affirmed and negated components. The basic logical form of modal statements relies on three main components, namely: one modal operator, one quantifier, and one predicative expression. Given that these components may be either affirmed or denied, let ± be a general operator symbolizing either the affirmation or negation of components. Assuming the validity of the following classical equivalences,

$$A \to B \equiv \neg(A \wedge \neg B)$$
$$\neg\neg A \equiv A$$
$$\forall x Ax \equiv \neg\exists x \neg Ax$$

the 32 modal statements may be rephrased as follows according to the two kinds of scope between quantifiers and modalities. (We are very grateful to Saloua Chatti for her valuable comments on the logical form of determinate (non-contingent) and contingent (indeterminate) statements; for one cannot distribute universal quantifiers over disjunctions and existential quantifiers over conjunctions, as was explained by [11]. However, note that there is a difference between two forms of undue distribution: in a non-modal context, where the tautological $\forall x(Px \vee \neg Px)$ does not entail $\forall x Px \vee \forall x \neg P$; in a modal statement with internal negation (as the case is herein), where the non-tautological $\forall x(\Box Px \vee \Box \neg Px)$ does not entail $\forall x \Box Px \vee \forall x \Box \neg Px$.)

<div align="center">

De dicto modal statements:
$$\pm\Diamond\pm\exists x(Sx \wedge \pm Px)$$

</div>

(i)     $\Box$A: $\neg\Diamond\exists x(Sx \wedge \neg Px) = \Box\forall x(Sx \to Px)$
(ii)    $\Box$E: $\neg\Diamond\exists x(Sx \wedge Px) = \Box\forall x(Sx \to \neg Px)$
(iii)   $\Box$I: $\neg\Diamond\neg\exists x(Sx \wedge Px) = \Box\exists x(Sx \wedge Px)$
(iv)    $\Box$O: $\neg\Diamond\neg\exists x(Sx \wedge \neg Px) = \Box\exists x(Sx \wedge \neg Px)$
(v)     $\Diamond$A: $\Diamond\neg\exists x(Sx \wedge \neg Px) = \Diamond\forall x(Sx \to Px)$
(vi)    $\Diamond$E: $\Diamond\neg\exists x(Sx \wedge Px) = \Diamond\forall x(Sx \to \neg Px)$
(vii)   $\Diamond$I: $\Diamond\exists x(Sx \wedge Px)$
(viii)  $\Diamond$O: $\Diamond\exists x(Sx \wedge \neg Px)$
(ix)    $\Delta$A: $\Box\forall x(Sx \to Px) \vee \Box\exists x(Sx \wedge \neg Px)$
(x)     $\Delta$E: $\Box\forall x(Sx \to \neg Px) \vee \Box\exists x(Sx \wedge Px)$
(xi)    $\Delta$I: $\Box\exists x(Sx \wedge Px) \vee \Box\forall x(Sx \to \neg Px)$
(xii)   $\Delta$O: $\Box\exists x(Sx \wedge \neg Px) \vee \Box\forall x(Sx \to Px)$
(xiii)  $\nabla$A: $\Diamond\forall x(Sx \to Px) \wedge \Diamond\exists x(Sx \wedge \neg Px)$
(xiv)   $\nabla$E: $\Diamond\forall x(Sx \to \neg Px) \wedge \Diamond\exists x(Sx \wedge Px)$
(xv)    $\nabla$I: $\Diamond\exists x(Sx \wedge Px) \wedge \Diamond\forall x(Sx \to \neg Px)$
(xvi)   $\nabla$O: $\Diamond\exists x(Sx \wedge \neg Px) \wedge \Diamond\forall x(Sx \to Px)$

<div align="center">

De re modal statements:
$$\pm\exists x(Sx \wedge \pm\Diamond\pm Px)$$

</div>

(xvii)   A$\Box$: $\forall x(Sx \to \Box Px)$
(xviii)  A$\Diamond$: $\forall x(Sx \to \Diamond Px)$
(xix)    A$\Delta$: $\forall x((Sx \to \Box Px) \vee (Sx \to \Box\neg Px))$
(xx)     A$\nabla$: $\forall x((Sx \to \Diamond Px) \wedge (Sx \to \Diamond\neg Px))$
(xxi)    E$\Box$: $\forall x(Sx \to \Box\neg Px)$
(xxii)   E$\Diamond$: $\forall x(Sx \to \Diamond\neg Px)$
(xxiii)  E$\Delta$: $\forall x((Sx \to \Box\neg Px) \vee (Sx \to \Box Px))$
(xxiv)   E$\nabla$: $\forall x((Sx \to \Diamond\neg Px) \wedge (Sx \to \Diamond Px))$
(xxv)    I$\Box$: $(\exists x)(Sx \wedge \Box Px)$

(xxvi)   I◇: $\exists x(Sx \wedge \Diamond Px)$
(xxvii)  I∆: $\exists x((Sx \wedge \Box Px) \vee (Sx \wedge \Box \neg Px))$
(xxviii) I∇: $\exists x((Sx \wedge \Diamond Px) \wedge (Sx \wedge \Diamond \neg Px))$
(xxix)   O□: $(\exists x)(Sx \wedge \Box \neg Px)$
(xxx)    O◇: $\exists x(Sx \wedge \Diamond \neg Px)$
(xxxi)   O∆: $\exists x((Sx \wedge \neg \Box Px) \vee (Sx \rightarrow \Box Px))$
(xxxii)  O∇: $\exists x((Sx \wedge \Diamond \neg Px) \wedge (Sx \wedge \Diamond Px))$

Note that the above renderings of contingency and determinacy differ from Ockham's versions, as we already mentioned at the end of Section 3. Although Ockham is right to apply De Morgan's rules by turning contingent formulas into determinate ones, we claim that he was wrong in his initial reading of contingency (whether universal or particular). The difference lies in the way in which negation occurs in the second conjuncts of his universal and particular de contingenti. Indeed, contingency means that it is possible for a given statement and its negation to be true. Now consider the statement ∇A: "It is contingent that every man is an animal", where contingency applies to the universal affirmative A. We already saw that, according to Ockham, the latter means that it is possible that every man is an animal, and it is possible that no man is an animal (universal de contingenti). This kind of interpretation makes an *internal* use of negation in its second conjunct, "No man is an animal" (i.e., "Every man is *not* an animal"), whereas our previous definition of contingency means that, according to ∇A, it is possible that every man is an animal, and it is possible that *not* every man is an animal, i.e., it is possible that some man is not an animal. In other words, our definition of determinacy and contingency makes an *external* use of negation in its second conjunct.

In order to reflect this discrepancy with Ockham, the following proposes a systematic way to account for the modal statements (i)–(xxxii) and their mutual logical relations. Our proposal is now to establish the differences in meaning in terms of truth-conditions: assuming a truth-conditional view of meaning, the point is to show that any of the above modal statements mean the same whenever they share the same truth-conditions. How can we account for these conditions in modal logic, recalling that modal operators are non-truth-functional? Let us consider in the following an alternative kind of relational semantics for this purpose.

*4.2. Semantics*

Possible world semantics (or relational semantics) is the standard way to afford the truth-conditions of modal statements, particularly because Kripke's models help to capture the plural meaning of necessity and possibility in modal frames and their various accessibility relations between worlds. Instead of following that path, however, the next sections intend to explicate Ockham's view of modalities by means of a special relational semantics, numbering semantics, in which the meaning of a statement relies upon a partition of logical space into ordered sets of numbers. We assume in the following that necessity is treated as an S5 modality, and the point is to determine all logical interrelations between any modal statements. This results in an updated theory of opposition for (i)–(xxxii), with the help of a set-theoretical algebra to redefine the variety of logical relations between arbitrary formulas.

### 4.2.1. Relational Statements

First, let us rephrase the logical form of modal statements in order to make sense of the modal operators. Following the modern view of necessity as truth in all possible worlds, another way to say that is by claiming that, for example, every $S$ is necessarily $P$ if, and only if, $S$ is $P$ at every given world. Let $P$ be a dyadic relation between an individual and a world, such that $Paw$ reads "(the individual) $a$ is $P$ at $w$". Then the de dicto and de re modal statements can be rephrased into these new logical forms of second-order logic by quantifying over possible worlds:

De dicto modal statements:

$$\pm \exists w \pm \exists x \pm Pxw$$

De re modal statements:
$$\pm \exists x \pm \exists w \pm Pxw$$

The above reformulation of modal statements clearly shows that de re modal statements merely switch the ordering of quantifiers as they occur in their de dicto counterparts, recalling that modality is viewed now as a second kind of quantifier ranging over worlds. Apart from ontological scruples regarding what entities may occur in a world, a purely logical approach to the matter allows us to think of a relational statement like "*a* is *P* at *w*" (about being at) as a relational expression that is on a par with "*a* loves *b*" (about loving). Thus "being at" and "loving" are two equally dyadic predicates.

### 4.2.2. Relational Semantics

Once that analogy is admitted, we can construct a model for modal statements in which a world includes three kinds of entities: properties, individuals, and worlds. This means that a world may include another world as an element. Besides that, a minimum number of two individual values $a, b$ is required in order to make a difference between worlds at which everyone is $P$ and someone (but not everyone) is $P$. Given that the modal statements (i)–(xxxii) leave the predicate expression $Sx$ unchanged, models need not include a second property $S$ and may include only $P$ to make sense of these modal statements. (Of course, one can conceive a situation in which something is not $S$; however, this requires another, more complex logical form of modal statements in which the subject term $S$ can be either affirmed or negated. See the conclusions regarding this prospect.)

A minimal set for modal statements includes two individuals $a, b$ and (at least) one property $P$, together with two possible worlds $w_1, w_2$. The truth-value of a modal statement consists of knowing which individuals satisfy the property $P$ in which possible world, accordingly. Let us number the resulting 16 models, whose cardinal results from the powerset of $n = 4$ elements (2 individual values, 2 possible worlds):

$1 = \{Paw_1, Paw_2, Pbw_1, Pbw_2\}$
$2 = \{Paw_1, Paw_2, Pbw_1, \neg Pbw_2\}$
$3 = \{Paw_1, Paw_2, \neg Pbw_1, Pbw_2\}$
$4 = \{Paw_1, \neg Paw_2, Pbw_1, Pbw_2\}$
$5 = \{\neg Paw_1, Paw_2, Pbw_1, Pbw_2\}$
$6 = \{Paw_1, Paw_2, \neg Pbw_1, \neg Pbw_2\}$
$7 = \{Paw_1, \neg Paw_2, Pbw_1, \neg Pbw_2\}$
$8 = \{\neg Paw_1, Paw_2, Pbw_1, \neg Pbw_2\}$
$9 = \{Paw_1, \neg Paw_2, \neg Pbw_1, Pbw_2\}$
$10 = \{\neg Paw_1, Paw_2, \neg Pbw_1, Pbw_2\}$
$11 = \{\neg Paw_1, \neg Paw_2, Pbw_1, Pbw_2\}$
$12 = \{Paw_1, \neg Paw_2, \neg Pbw_1, \neg Pbw_2\}$
$13 = \{\neg Paw_1, Paw_2, \neg Pbw_1, \neg Pbw_2\}$
$14 = \{\neg Paw_1, \neg Paw_2, Pbw_1, \neg Pbw_2\}$
$15 = \{\neg Paw_1, \neg Paw_2, \neg Pbw_1, Pbw_2\}$
$16 = \{\neg Paw_1, \neg Paw_2, \neg Pbw_1, \neg Pbw_2\}$

It can be shown that each of the above 16 sets of propositional sentences belongs or does not belong to the truth-conditions of Ockham's modal statements. To determine which ones characterize each of the corresponding 32 formulas is the task of the following section.

### 4.2.3. Numbering Semantics

The truth-value of any modal statement $X$ can be codified in terms of a *model set*, i.e., a set of the set of propositional sentences that are satisfied by it. In order to simplify the ensuing semantic representation of formulas, a developed technique has been implemented elsewhere (see [12–14]) to depict these sets in terms of ordered Boolean bits, 1 or 0. The difference between this technique and the following model sets is that a Boolean bit

corresponds to a unique kind of model, including more than one model, whereas we are now going to depict each of the single models (for the sake of pedagogical clarity). Therefore, the following numbering semantics corresponds to a more descriptive semantics in which every single number refers to a single model among the 16.

Accordingly, let us symbolize by $\mathcal{W}(X)$ the set of numbers (from 1 to 16) that corresponds to the models satisfied by the formula $X$.

The set-theoretical import of that semantics entails that conjunction and disjunction between statements are rendered as the intersection and union of their corresponding numberings, accordingly. Given that modal statements have been understood as mixed quantified statements, it is no surprise that a number of them are equivalent to each other, i.e., have the same characteristic numbering. Indeed, any statement including two quantifiers of the same sort (universal or particular) is equivalent with its switched counterpart so that, e.g., the de dicto statement (i), "It is necessary that every $S$ is $P$" means the same as its de re counterpart (xvii), "Every $S$ is necessarily $P$". This appears in the following list of the characteristic numberings of modal statements, including only 20 characteristic sets for a total of 32 formulas (see Appendix A for a constructive proof of these ordered numbers):

(1)  $\mathcal{W}(\Box A) = \mathcal{W}(A\Box) = \{1\}$
(2)  $\mathcal{W}(\Box E) = \mathcal{W}(E\Box) = \{16\}$
(3)  $\mathcal{W}(\Box I) = \{1, 2, 3, 4, 5, 6, 8, 9, 11\}$
(4)  $\mathcal{W}(\Box O) = \{6, 8, 9, 11, 12, 13, 14, 15, 16\}$
(5)  $\mathcal{W}(\Diamond A) = \{1, 2, 3, 4, 5, 7, 10\}$
(6)  $\mathcal{W}(\Diamond E) = \{7, 10, 12, 13, 14, 15, 16\}$
(7)  $\mathcal{W}(\Diamond I) = \mathcal{W}(I\Diamond) = \{1, 2, 3, 4, 5, 6, 7, 8, 9, 10, 11, 12, 13, 14, 15\}$
(8)  $\mathcal{W}(\Diamond O) = \mathcal{W}(O\Diamond) = \{2, 3, 4, 5, 6, 7, 8, 9, 10, 11, 12, 13, 14, 15, 16\}$
(9)  $\mathcal{W}(\Delta A) = \mathcal{W}(\Delta O) = \{1, 6, 8, 9, 11, 12, 13, 14, 15, 16\}$
(10) $\mathcal{W}(\Delta E) = \mathcal{W}(\Delta I) = \{1, 2, 3, 4, 5, 6, 8, 9, 11, 16\}$
(11) $\mathcal{W}(\nabla A) = \mathcal{W}(\nabla O) = \{2, 3, 4, 5, 7, 10\}$
(12) $\mathcal{W}(\nabla E) = \mathcal{W}(\nabla I) = \{7, 10, 12, 13, 14, 15\}$
(13) $\mathcal{W}(A\Diamond) = \{1, 2, 3, 4, 5, 7, 8, 9, 10\}$
(14) $\mathcal{W}(A\Delta) = \mathcal{W}(E\Delta) = \{1, 6, 11, 16\}$
(15) $\mathcal{W}(A\nabla) = \mathcal{W}(E\nabla) = \{7, 8, 9, 10\}$
(16) $\mathcal{W}(E\Diamond) = \{7, 8, 9, 10, 12, 13, 14, 15, 16\}$
(17) $\mathcal{W}(I\Box) = \{1, 2, 3, 4, 5, 6, 11\}$
(18) $\mathcal{W}(I\Delta) = \mathcal{W}(O\Delta) = \{1, 2, 3, 4, 5, 6, 11, 12, 13, 14, 15, 16\}$
(19) $\mathcal{W}(I\nabla) = \mathcal{W}(O\nabla) = \{2, 3, 4, 5, 7, 8, 9, 10, 12, 13, 14, 15\}$
(20) $\mathcal{W}(O\Box) = \{6, 11, 12, 13, 14, 15, 16\}$

It is worthwhile to note two things in the above numberings.

First, all these do *not* include Ockham's four de dicto contingent and determinate statements: $U$, $Y$, $K$, and $Z$. This is, again, because we take his de contingenti formulas to rely on a wrong occurrence of internal negation (see the end of Sections 3 and 4.1). Thus, there is a logical difference between $\Delta I$ and Ockham's $U = (1) \vee (2)$ and between $\Delta A$ and Ockham's $K = (3) \vee (4)$; the same consequently holds for their contradictories, i.e., respectively, Ockham's $Y = (7) \wedge (8)$ and $Z = (5)/(13) \wedge (6)/(16)$. In the case of Z, Ockham considers (5) equivalent to (13) and (6) equivalent to (16), which is a mistake due to his content-based account of de dicto and de re. See Appendix A for a proof of their differences in terms of truth-conditions.

Second, the above equivalences match with the famous Barcan formulas by accepting the following equivalences:

$$\forall x \Box F x \leftrightarrow \Box \forall x F x$$
$$\exists x \Diamond F x \leftrightarrow \Diamond \exists x F x$$

An expected objection to the above equivalences is that an individual may exist in a possible world without existing in the actual world, thus invalidating the entailment relation from

$\Diamond \exists x Fx$ to $\exists x \Diamond Fx$. Although the main reason for this equivalence in our semantics is that no clear-cut distinction is made between possible worlds and the actual world, a theoretical reason may be advanced to defend it as well (see [15]): if there is a world at which an object, say $a$, is $F$, so $a$ may be $F$ in the actual world without being so after all. Our model set assumes a set of constant individuals, such that whatever exists in a world also exists in all the other ones. However, even in the contrary case, it seems that the Barcan formulas would still hold because a given property $F$ may be satisfied by one individual *whichever*: if there is a world at which something is $F$, so something is $F$ in the actual world without requiring that it is one and the same individual in both cases, for how can we individuate an object without specifying its properties? This philosophical issue is left open in the present paper, and our point is just to claim that the above logical equivalences cannot be taken to be counterintuitive without some special philosophical assumptions.

### 4.2.4. Logical Relations

Finally, the characteristic numbering of modal statements can be used to identify the logical relations between any pair of them. For this purpose, we use the basic notions of model and counter-model. For any integer $x \in \{1–16\}$, we say that $x$ is a *model* of the formula $X$ *iff* $x \in \mathcal{W}(X)$, and $x$ is a *counter-model* of the formula $X$ *iff* $x \notin \mathcal{W}(X)$. Then the logical relations between formulas deal with models that any two formulas can share or not. For any two formulas $X, Y$, the fact that they are *compatible* means that they can share the same model; if, on the contrary, they are *incompatible*, this means that they cannot share any model or, in other words, that any model (or counter-model) of the first formula $X$ is a counter-model (or model) of the second formula $Y$. This results in the well-known set of four Aristotelian relations of opposition, including two cases of compatibility and one pattern of the entailment relation (viz. subalternation). Thus, for any related statements $X, Y$:

$X$ and $Y$ are *contraries* (symbol: *ct*) *iff* every model of $X$ is a counter-model of $Y$, but not every counter-model of $X$ is a model of $Y$.
$X$ and $Y$ are *contradictories* (symbol: *cd*) *iff* every model of $X$ is a counter-model of $Y$, and every counter-model of $X$ is a model of $Y$.
$X$ and $Y$ are *subcontraries* (symbol: *sct*) *iff* every counter-model of $X$ is a model of $Y$, but not every model of $X$ is a counter-model of $Y$.
$Y$ is *subaltern* to $X$ (symbol: *sb*) *iff* every model of $X$ is a model of $Y$, and every counter-model of $Y$ is a counter-model of $X$.
Finally, $X$ and $Y$ are *independent* from each other (symbols: *ind*) whenever they do not satisfy any of the above conditions.

Following the specialized literature on logical oppositions, a way to depict these numerous relations is by representing these in a logical diagram inspired by the traditional theory of oppositions. Let us see what happens with Ockham's modalities, and how a complete combination of their de dicto and de re modalities can be represented accordingly.

### 4.2.5. Increasing Diagrams

It has already been mentioned in a previous section that the Aristotelian square included a modal version of categorical propositions, and the extension of necessity and possibility to contingency naturally led to Blanché's hexagon. In addition, two other diagrammatic versions of modal logic were implemented under the impetus of works around other logicians: on the one hand, ref. [16] noticed that the history of logic contained a logical octagon of quantified modal logic behind the work of Buridan, and ref. [17] provided a formal semantics for it; on the other hand, ref. [18] devised a further dodecagon in showing that Avicenna proposed a set of logical relations between 12 statements.

In light of the preceding, one can surmise a closure of this increasing extension from the square onward: there can be many more than 12 modal statements in their de re–de dicto versions. A first representation (Figure 5) of modal statements is a logical dodecagon of strictly de dicto modal statements, where modal operators occur externally

and augment the four Aristotelian relations (contrariety, contradictoriness, subcontrariety, and subalternation) with one additional relation of independence. For sake of simplicity, the nature of the numerous logical relations between vertices is not displayed in the following three figures, and the reader is invited to check these in Appendix B.

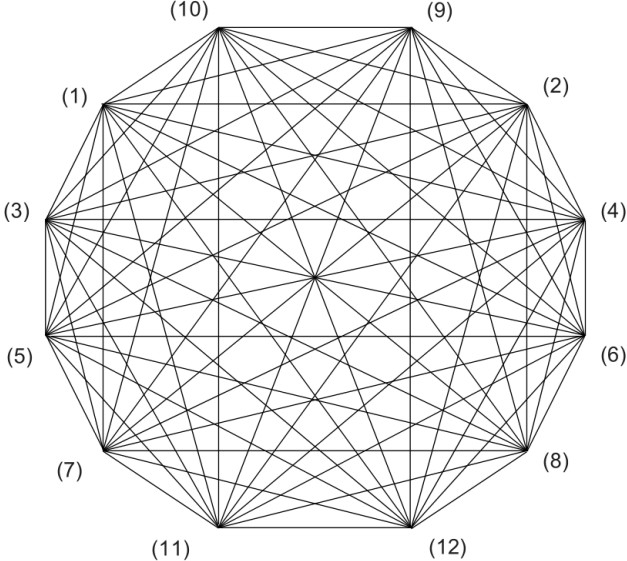

**Figure 5.** A first representation of modal statements.

A second representation (Figure 6) is a logical dodecagon of strictly de re modal statements, where modal operators occur internally only.

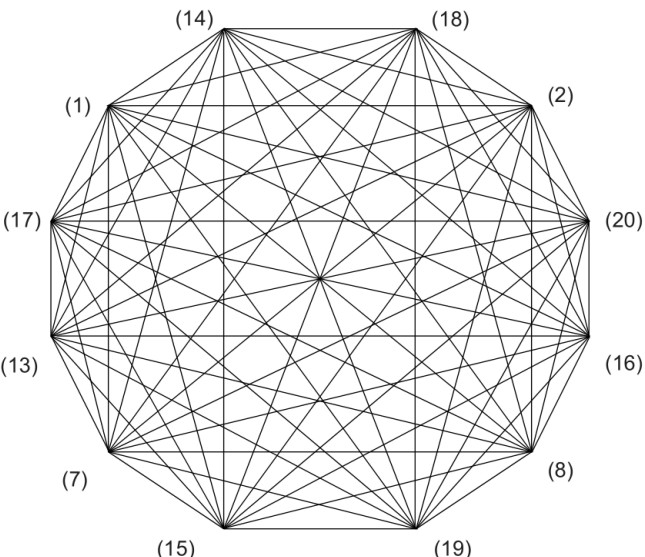

**Figure 6.** A second representation of modal statements.

Finally, a combination of all logical relations between de dicto and de re statements results in a set of a set of 20 formulas (Figure 7), that is, a new logical icosagon of modal statements.

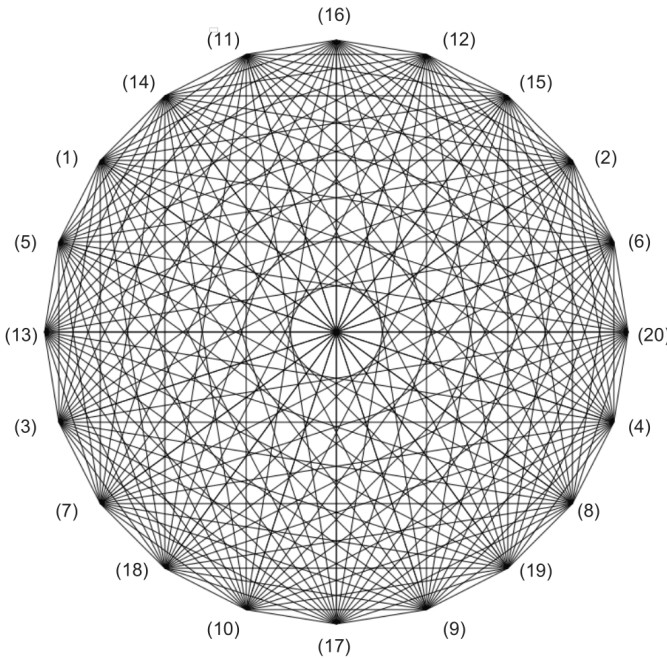

**Figure 7.** A new logical icosagon of modal statements.

The above diagram extends the previous 2 figures into an embedding icosagon that includes 27 logical squares (i.e., embedded structures with 1 relation of contrariety, 1 relation of subcontrariety, 2 relations of contradictoriness, and 2 relations of subalternation; see Appendix B for a complete list of these embedded logical squares), assuming that each vertex of that structure corresponds to single modal statements (10)–(20) and that the structure thereby excludes Ockham's disjunctive and conjunctive forms $U, Y, K, Z$.

Note finally that each of the implication relations between both de dicto and de re modal statements (see Section 3) is established in a set-theoretical way: each antecedent is a superaltern of its consequent so that, e.g., the first implication

$$\Box \forall x (Sx \rightarrow Px) \rightarrow \Diamond \forall x (Sx \rightarrow Px)$$

is a more customary way to claim that (1) entails (5) because the latter is subaltern to the former. See the above exhaustive table of logical relations between (1)–(16).

## 5. Conclusions

We have provided a survey of Ockham's theory of modal statements based on a de re–de dicto distinction in the use of modalities, and we analyzed in detail the cases of contingency and determinacy (or non-contingency) in order to extend Ockham's modal squares. Then we proposed a reformulation of modal statements in a systematic way by means of both a second-order translation and a corresponding relational semantics where statements were codified by numbering. Finally, we showed that Ockham's modal statements should be reformulated considering an external application of negation and hence reduced to an exhaustive set of 16 formulas, thereby leading to a comprehensive icosagon that encompasses the previous extensions from the Aristotelian square to Ockham's hexagon and Buridan's octagon of logical relations between these modal statements.

Let us recall that such a semantics relies on a special interpretation of necessity as truth in all possible worlds, i.e., models where the accessibility relation is an equivalence relation in terms of Kripke semantics. We favored an alternative semantics of ordered model sets, however, in order to construct an algebraic theory of logical relations between matching formulas (i.e., sharing the same logical form; see [19]). An interesting development in the proposed numbering semantics consists of constructing model sets for non-equivalent accessibility relations, thus applying to temporal, epistemic, or deontic interpretations of

the modal operators. This project, however, goes beyond our present purpose to make sense of Ockham's proper theory of modalities with modern formal tools. At any rate, such a special semantics has already been developed in other separate works and Boolean ways (see e.g., [12,14]) and could turn out to be a nice trade-off between Kripke semantics and the prior algebraic tradition of modal logic.

Another prospect is to extend the logical form of Ockham's modal statements by also negating the subject term *S* of the categorical statements A, E, I, O. Such an extension was devised by previous logicians (see especially [20]), and this amounts to proposing a modal version of Keynesian categorical propositions (wherein *S* is always negated), whereas Ockham adhered to a modal version of Aristotelian categorical propositions (wherein *S* is always affirmed). A bitstring semantics for both Aristotelian and Keynesian categorical propositions has been recently set forth (see [21]). Its modal de re and de dicto versions remain to be generated by applying modal operators to logical forms like

$$\pm\Diamond\pm\exists x(\pm Sx \wedge \pm Px),$$
$$\text{or}$$
$$\pm\exists x\pm\Diamond(\pm Sx \wedge \pm Px).$$

**Author Contributions:** Writing—original draft, D.F. and F.S. All authors have read and agreed to the published version of the manuscript.

**Funding:** The present article has been published thanks to funding provided by the Swiss National Science Foundation. SNSF Grant ID: 20074.

**Conflicts of Interest:** The authors declare no conflict of interest.

## Appendix A. Numbering Semantics for De dicto and De re Modal Statements

For any binary connective $\circ = \{\wedge, \vee\}$, Ockham's modal operators can be read in terms of conjunct or disjunct truths in possible worlds. Recalling the aforementioned model that includes two individuals $a, b$ and two possible worlds $w_1, w_2$, a formula like 'Every *S* is necessarily *P*' means that every *S* is *P* in both $w_1$ and $w_2$, whereas 'Every *S* is possibly *P*' means that every *S* is *P* in $w_1$ or in $w_2$ (or both). Let us recall that *S* is assumed to be non-empty in Ockam's modal theory, so that the difference between de dicto and de re modal statements lies in the contents of any main conjunct or disjunct: when de dicto, these include both individuals $a, b$ and only one possible world and when de re, these include both possible worlds and only one individual.

De dicto modal statements: $(\pm Paw_1 \circ \pm Pbw_1) \circ (\pm Paw_2 \circ \pm Pbw_2)$

Every de dicto statement amounts to a combination of 8 subformulas ①–⑧, viz., conjunctions (necessary statements) or disjunctions (possible statements) about what $a$ and $b$ are in a single world $w_1$ or $w_2$. That is:

①  $= (Paw_1 \wedge Pbw_1) = \{1, 2, 4, 7\}$
②  $= (Paw_2 \wedge Pbw_2) = \{1, 3, 5, 10\}$
③  $= (Paw_1 \vee Pbw_1) = \{1, 2, 3, 4, 5, 6, 7, 8, 9, 11, 12, 14\}$
④  $= (Paw_2 \vee Pbw_2) = \{1, 2, 3, 4, 5, 6, 8, 9, 10, 11, 13, 15\}$
⑤  $= (\neg Paw_1 \wedge \neg Pbw_1) = \{10, 13, 15, 16\}$
⑥  $= (\neg Paw_2 \wedge \neg Pbw_2) = \{7, 12, 14, 16\}$
⑦  $= (\neg Paw_1 \vee \neg Pbw_1) = \{3, 5, 6, 8, 9, 10, 11, 12, 13, 14, 15, 16\}$
⑧  $= (\neg Paw_2 \vee \neg Pbw_2) = \{2, 4, 6, 7, 8, 9, 11, 12, 13, 14, 15, 16\}$

Every de dicto statement consists of combining either of these subformulas with either conjunction or disjunction, depending on which modal operator occurs in a broad scope: conjunction, with necessity; disjunction, with possibility; disjunction of conjuncts, with determinacy (non-contingency); conjunction of disjuncts, with contingency (indeterminacy).

$\Box \mathcal{Q} = \bigcirc \wedge \bigcirc$
$\mathcal{W}(\Box\text{A}) = \mathcal{W}(①) \wedge ② = \{1, 2, 4, 7\} \cap \{1, 3, 5, 10\} = \{1\}$

$\mathcal{W}(\square E) = \mathcal{W}(⑤ \wedge ⑥) = \{10, 13, 15, 16\} \cap \{7, 12, 14, 16\} = \{16\}$

$\mathcal{W}(\square I) = \mathcal{W}(③ \wedge ④) = \{1, 2, 3, 4, 5, 6, 7, 8, 9, 11, 12, 14\} \cap \{1, 2, 3, 4, 5, 6, 8, 9, 10, 11, 13, 15\} = \{1, 2, 3, 4, 5, 6, 8, 9, 11\}$

$\mathcal{W}(\square O) = \mathcal{W}(⑦ \wedge ⑧) = \{3, 5, 6, 8, 9, 10, 11, 12, 13, 14, 15, 16\} \cap \{2, 4, 6, 7, 8, 9, 11, 12, 13, 14, 15, 16\} = \{6, 8, 9, 11, 12, 13, 14, 15, 16\}$

$\lozenge \mathcal{Q} = \bigcirc \vee \bigcirc$

$\mathcal{W}(\lozenge A) = \mathcal{W}(① \vee ②) = \{1, 2, 4, 7\} \cup \{1, 3, 5, 10\} = \{1, 2, 3, 4, 5, 7, 10\}$

$\mathcal{W}(\lozenge E) = \mathcal{W}(⑤ \vee ⑥) = \{10, 13, 15, 16\} \cup \{7, 12, 14, 16\} = \{7, 10, 12, 13, 14, 15, 16\}$

$\mathcal{W}(\lozenge I) = \mathcal{W}(③ \vee ④) = \{1, 2, 3, 4, 5, 6, 7, 8, 9, 11, 12, 14\} \cup \{1, 2, 3, 4, 5, 6, 8, 9, 10, 11, 13, 15\} = \{1, 2, 3, 4, 5, 6, 7, 8, 9, 10, 11, 12, 13, 14, 15\}$

$\mathcal{W}(\lozenge O) = \mathcal{W}(⑦ \vee ⑧) = \{3, 5, 6, 8, 9, 10, 11, 12, 13, 14, 15, 16\} \cup \{2, 4, 6, 7, 8, 9, 11, 12, 13, 14, 15, 16\} = \{2, 3, 4, 5, 6, 7, 8, 9, 10, 11, 12, 13, 14, 15, 16\}$

$\Delta \mathcal{Q} = (\bigcirc \wedge \bigcirc) \vee (\bigcirc \wedge \bigcirc)$

$\mathcal{W}(\Delta A) = \mathcal{W}((① \wedge ②) \vee (⑦ \wedge ⑧)) = \{1\} \cup \{6, 8, 9, 11, 12, 13, 14, 15, 16\} = \{1, 6, 8, 9, 11, 12, 13, 14, 15, 16\}$

$\mathcal{W}(\Delta E) = \mathcal{W}((⑤ \wedge ⑥) \vee (③ \wedge ④)) = \{16\} \cup \{1, 2, 3, 4, 5, 6, 8, 9, 11\} = \{1, 2, 3, 4, 5, 6, 8, 9, 11, 16\}$

$\mathcal{W}(\Delta I) = \mathcal{W}((③ \wedge ④) \vee (⑤ \wedge ⑥)) = \{1, 2, 3, 4, 5, 6, 8, 9, 11\} \cup \{16\} = \{1, 2, 3, 4, 5, 6, 8, 9, 11, 16\}$

$\mathcal{W}(\Delta O) = \mathcal{W}((⑦ \wedge ⑧) \vee (① \wedge ②)) = \{6, 8, 9, 11, 12, 13, 14, 15, 16\} \cup \{1\} = \{1, 6, 8, 9, 11, 12, 13, 14, 15, 16\}$

$\nabla \mathcal{Q} = (\bigcirc \vee \bigcirc) \wedge (\bigcirc \vee \bigcirc)$

$\mathcal{W}(\nabla A) = \mathcal{W}((① \vee ②) \wedge (⑦ \vee ⑧)) = \{1, 2, 3, 4, 5, 7, 10\} \cap \{2, 3, 4, 5, 6, 7, 8, 9, 10, 11, 12, 13, 14, 15, 16\} = \{2, 3, 4, 5, 7, 10\}$

$\mathcal{W}(\nabla E) = \mathcal{W}((⑤ \vee ⑥) \wedge (③ \vee ④)) = \{7, 10, 12, 13, 14, 15, 16\} \cap \{1, 2, 3, 4, 5, 6, 7, 8, 9, 10, 11, 12, 13, 14, 15\} = \{7, 10, 12, 13, 14, 15\}$

$\mathcal{W}(\nabla I) = \mathcal{W}((③ \vee ④) \wedge (⑤ \vee ⑥)) = \{1, 2, 3, 4, 5, 6, 7, 8, 9, 10, 11, 12, \&3, 14, 15\} \cap \{7, 10, 12, 13, 14, 15, 16\} = \{7, 10, 12, 13, 14, 15\}$

$\mathcal{W}(\nabla O) = \mathcal{W}(\mathcal{W}((⑦ \vee ⑧) \wedge (① \vee ②)) = \{2, 3, 4, 5, 6, 7, 8, 9, 10, 11, 12, 13, 14, 15, 16\} \cap \{1, 2, 3, 4, 5, 7, 10\} = \{2, 3, 4, 5, 7, 10\}$

De re modal statements: $(\pm Paw_1 \circ \pm Paw_2) \circ (\pm Pbw_1 \circ \pm Pbw_2)$

Every de re statement amounts to a combination of 8 other subformulas ⑨–⑯, viz., conjunctions (necessary statements) or disjunctions (possible statements) about what *a* or *b* is in both worlds $w_1$ and $w_2$. That is:

⑨ $= (Paw_1 \wedge Paw_2) = \{1, 2, 3, 6\}$

⑩ $= (Pbw_1 \vee Pbw_2) = \{1, 4, 5, 11\}$

⑪ $= (Paw_1 \wedge Paw_2) = \{1, 2, 3, 4, 5, 6, 7, 8, 9, 10, 12, 13\}$

⑫ $= (Pbw_1 \vee Pbw_2) = \{1, 2, 3, 4, 5, 7, 8, 9, 10, 11, 14, 15\}$

⑬ $= (\neg Paw_1 \wedge \neg Paw_2) = \{11, 14, 15, 16\}$

⑭ $= (\neg Pbw_1 \vee \neg Pbw_2) = \{6, 12, 13, 16\}$

⑮ $= (\neg Paw_1 \wedge \neg Paw_2) = \{4, 5, 7, 8, 9, 10, 11, 12, 13, 14, 15, 16\}$

⑯ $= (\neg Pbw_1 \vee \neg Pbw_2) = \{2, 3, 6, 7, 8, 9, 10, 12, 13, 14, 15, 16\}$

Every de re statement consists of combining either of these subformulas with either conjunction or disjunction, depending on which quantifying operator occurs in a broad scope: conjunction, with universal quantifiers; disjunction, with existential quantifiers; disjunction of conjuncts, when existential quantifiers apply to determinacy (non-contingency); conjunction of disjuncts, when universal quantifiers apply to contingency (indeterminacy).

$A\mathcal{M} = (\bigcirc \circ \bigcirc) \wedge (\bigcirc \circ \bigcirc)$

$\mathcal{W}(A\square) = \mathcal{W}(⑨ \wedge ⑩) = \{1, 2, 3, 6\} \wedge \{1, 4, 5, 11\} = \{1\}$

$\mathcal{W}(A\lozenge) = \mathcal{W}(⑪ \wedge ⑫) = \{1, 2, 3, 4, 5, 6, 7, 8, 9, 10, 12, 13\} \cap \{1, 2, 3, 4, 5, 7, 8, 9, 10, 11, 14, 15\} = \{1, 2, 3, 4, 5, 7, 8, 9, 10\}$

$\mathcal{W}(A\Delta) = \mathcal{W}(⑨ \wedge ⑬) \vee (⑩ \wedge ⑭)) = (\{1, 2, 3, 6\} \cup \{11, 14, 15, 16\}) \cap (\{1, 4, 5, 11\} \cup \{6, 12, 13, 16\}) = \{1, 2, 3, 6, 11, 14, 15, 16\} \cap \{1, 4, 5, 6, 11, 12, 13, 16\} = \{1, 6, 11, 16\}$

$\mathcal{W}(A\nabla) = \mathcal{W}(⑪ \wedge ⑮) \wedge (⑫ \wedge ⑯) = (\{1,2,3,4,5,6,7,8,9,10,12,13\} \cap \{4,5,7,8,9,10,11,12,13,14,15,16\}) \cap (\{1,2,3,4,5,7,8,9,10,14\} \cap \{2,3,6,7,8,9,10,12,13,14,15,16\}) = \{4,5,7,8,9,10,12,13\} \cap \{2,3,7,8,9,10,14,15\} = \{7,8,9,10\}$

$E\mathcal{M} = (\bigcirc \circ \bigcirc) \wedge (\bigcirc \circ \bigcirc)$

$\mathcal{W}(E\square) = \mathcal{W}(⑬ \wedge ⑭) = \{11,14,15,16\} \cap \{6,12,13,16\} = \{16\}$

$\mathcal{W}(E\lozenge) = \mathcal{W}(⑮ \wedge ⑯) = \{4,5,7,8,9,10,11,12,13,14,15,16\} \cap \{2,3,6,7,8,9,10,12,13,14,15,16\} = \{7,8,9,10,12,13,14,15,16\}$

$\mathcal{W}(E\Delta) = \mathcal{W}((⑬ \vee ⑨) \wedge (⑭ \wedge ⑩)) = (\{11,14,15,16\} \cup \{1,2,3,6\}) \cap (\{6,12,13,16\} \cup \{1,4,5,11\}) = \{1,2,3,6,11,14,15,16\} \cap \{1,4,5,6,11,12,13,16\} = \{1,6,11,16\}$

$\mathcal{W}(E\nabla) = \mathcal{W}((⑮ \wedge ⑪) \wedge (⑯ \wedge ⑫)) = (\{4,5,7,8,9,10,11,12,13,14,15,16\} \cap \{1,2,3,4,5,6,7,8,9,10,12,13\}) \cap (\{2,3,6,7,8,9,10,12,13,14,15,16\} \cap \{1,2,3,4,5,7,8,9,10,14\}) = \{2,3,7,8,9,10,14,15\} \cap \{4,5,7,8,9,10,12,13\} = \{7,8,9,10\}$

$I\mathcal{M} = (\bigcirc \circ \bigcirc) \vee (\bigcirc \circ \bigcirc)$

$\mathcal{W}(I\square) = \mathcal{W}(⑨ \vee ⑩) = \{1,2,3,6\} \cup \{1,4,5,11\} = \{1,2,3,4,5,6,11\}$

$\mathcal{W}(I\lozenge) = \mathcal{W}(⑪ \vee ⑫) = \{1,2,3,4,5,6,7,8,9,10,12,13\} \cup \{1,2,3,4,5,7,8,9,10,11,14,15\} = \{1,2,3,4,5,6,7,8,9,10,11,12,13,14,15\}$

$\mathcal{W}(I\Delta) = \mathcal{W}((⑨ \vee ⑬) \vee (⑩ \vee ⑭)) = (\{1,2,3,6\} \cup \{11,14,15,16\}) \cup (\{1,4,5,11\} \cup \{6,12,13,16\}) = \{1,2,3,6,11,14,15,16\} \cup \{1,4,5,6,11,12,13,16\} = \{1,2,3,4,5,6,11,12,13,14,15,16\}$

$\mathcal{W}(I\nabla) = \mathcal{W}((⑪ \wedge ⑮) \vee (⑫ \wedge ⑯)) = (\{1,2,3,4,5,6,7,8,9,10,12,13\} \cap \{4,5,7,8,9,10,11,12,13,14,15,16\}) \cup (\{1,2,3,4,5,7,8,9,10,14\} \cap \{2,3,6,7,8,9,10,12,13,14,15,16\}) = \{4,5,7,8,9,10,12,13\} \cup \{2,3,7,8,9,10,14,15\} = \{2,3,4,7,8,9,10,12,13,14,15\}$

$O\mathcal{M} = (\bigcirc \circ \bigcirc) \vee (\bigcirc \circ \bigcirc)$

$\mathcal{W}(O\square) = \mathcal{W}(⑬ \vee ⑭) = \{11,14,15,16\} \cup \{6,12,13,16\} = \{6,11,12,13,14,15,16\}$

$\mathcal{W}(O\lozenge) = \mathcal{W}(⑮ \vee ⑯) = \{4,5,7,8,9,10,11,12,13,14,15,16\} \cup \{2,3,6,7,8,9,10,12,13,14,15,16\} = \{2,3,4,5,6,7,8,9,10,11,12,13,14,15,16\}$

$\mathcal{W}(O\Delta) = \mathcal{W}((⑬ \vee ⑨) \vee (⑭ \vee ⑩)) = (\{11,14,15,16\} \cup \{1,2,3,6\}) \cup (\{6,12,13,16\} \cup \{1,4,5,11\}) = \{1,2,3,6,11,14,15,16\} \cup \{1,4,5,6,11,12,13,16\} = \{1,2,3,4,5,6,11,12,13,14,15,16\}$

$\mathcal{W}(O\nabla) = \mathcal{W}((⑮ \wedge ⑪) \wedge (⑯ \wedge ⑫)) = (\{4,5,7,8,9,10,11,12,13,14,15,16\} \cap \{1,2,3,4,5,6,7,8,9,10,12,13\}) \cup (\{2,3,7,8,9,10,14,15\} \cap \{1,2,3,4,5,7,8,9,10,11,14,15\}) = \{4,5,7,8,9,10,12,13\} \cup \{2,3,7,8,9,14,15\} = \{2,3,4,5,7,8,9,10,12,13,14,15\}$

Finally, it can be proved that what Ockham takes to be contingent and determinate statements differs from our above translations of contingencies and determinacies.

Let $(X)_O$ symbolize Ockham's reading of these contingent and determinate modal statements. The difference between their set-theoretical meanings and ours clearly appears in their characteristic numberings:

$(\Delta A)_O = K = \square I \vee \square O$

$\mathcal{W}(\Delta A)_O = \mathcal{W}(\square I) \cup \mathcal{W}(\square O) = \{1,2,3,4,5,6,7,8,9,11,12,14\} \cup \{6,8,9,11,12,13,14,15,16\} = \{1,2,3,4,5,6,7,8,9,11,12,13,14,15,16\}$

$\mathcal{W}(\Delta A) = \{1,6,8,9,11,12,13,14,15,16\}$

Hence $(\Delta A)_O \neq \Delta A$.

$(\Delta I)_O = U = \square A \vee \square E$

$\mathcal{W}(\Delta I)_O = \mathcal{W}(\square A) \cup \mathcal{W}(\square E) = \{1\} \cup \{16\} = \{1,16\}$

$\mathcal{W}(\Delta I) = \{1,2,3,4,5,6,8,9,11,16\}$

Hence $(\Delta I)_O \neq \Delta I$.

$(\nabla A)_O = Z = \lozenge A \wedge \lozenge E$

$\mathcal{W}(\nabla A)_O = \mathcal{W}(\lozenge A) \cap \mathcal{W}(\lozenge E) = \{1,2,3,4,5,7,10\} \cap \{7,10,12,13,14,15,16\} = \{7,10\}$

$\mathcal{W}(\nabla A) = \{2,3,4,5,7,10\}$

Hence $(\nabla A)_O \neq \nabla A$.

$(\nabla I)_O = Y = \lozenge I \wedge \lozenge O$

$\mathcal{W}(\nabla I)_O = \mathcal{W}(\lozenge I) \cap \mathcal{W}(\lozenge O) = \{1,2,3,4,5,6,7,8,9,10,11,12,13,14,15\} \cap \{2,3,4,5,6,7,8,9,10,11,12,13,14,15,16\} = \{2,3,4,5,6,7,8,9,10,11,12,13,14,15\}$

$\mathcal{W}(\nabla I) = \{7, 10, 12, 13, 14, 15\}$
Hence $(\nabla I)_O \neq \nabla I$.

## Appendix B. Logical Relations between De dicto and De re Modal Statements

The set of 20 modal statements leads to a set of $20(20-1)/2 = 190$ logical interrelations as depicted in the following table in which each kind of logical relation is symbolized as follows: 'ct' for contrariety, 'cd' for contradiction, 'sct' for subcontrariety, 'sb' for subalternation, 'sp' for superalternation, and 'ind' for independence.

|      | (1) | (2) | (3) | (4) | (5) | (6) | (7) | (8) | (9) | (10) |
|------|-----|-----|-----|-----|-----|-----|-----|-----|-----|------|
| (1)  |     | ct  | sp  | ct  | sp  | ct  | sp  | cd  | sp  | sp   |
| (2)  | ct  |     | ct  | sp  | ct  | sp  | cd  | sp  | sp  | sp   |
| (3)  | sb  | ct  |     | ind | ind | cd  | sp  | sct | ind | sp   |
| (4)  | ct  | sb  | ind |     | cd  | ind | sct | sp  | sp  | ind  |
| (5)  | sb  | ct  | ind | cd  |     | ind | sp  | sct | sct | ind  |
| (6)  | ct  | sb  | cd  | ind | ind |     | sct | sp  | ind | sct  |
| (7)  | sb  | cd  | sb  | sct | sb  | sct |     | sct | sct | sct  |
| (8)  | cd  | sb  | sct | sb  | sct | sct | sct |     | sct | sct  |
| (9)  | sb  | sb  | ind | sb  | sct | ind | sct | sct |     | ind  |
| (10) | sb  | sb  | sb  | ind | ind | sct | sct | sct | ind |      |

|      | (11) | (12) | (13) | (14) | (15) | (16) | (17) | (18) | (19) | (20) |
|------|------|------|------|------|------|------|------|------|------|------|
| (11) |      | ind  | sp   | ct   | ind  | ind  | ind  | ind  | sp   | ct   |
| (12) | ind  |      | ind  | ct   | ind  | sp   | ct   | ind  | sp   | ind  |
| (13) | sb   | ind  |      | ind  | sb   | ind  | ind  | sct  | ind  | cd   |
| (14) | ct   | ct   | ind  |      | ct   | ind  | ind  | sp   | cd   | ind  |
| (15) | ind  | ind  | sp   | ct   |      | sp   | ct   | cd   | sp   | ct   |
| (16) | ind  | sb   | ind  | ind  | sb   |      | cd   | sct  | ind  | ind  |
| (17) | ind  | ct   | ind  | ind  | ct   | cd   |      | sp   | ind  | ind  |
| (18) | ind  | ind  | sct  | sb   | cd   | sct  | sb   |      | sct  | sb   |
| (19) | sb   | sb   | ind  | ind  | sb   | ind  | ind  | sct  |      | ind  |
| (20) | ct   | ind  | cd   | ind  | ct   | ind  | ind  | sp   | ind  |      |

|      | (11) | (12) | (13) | (14) | (15) | (16) | (17) | (18) | (19) | (20) |
|------|------|------|------|------|------|------|------|------|------|------|
| (1)  | ct   | ct   | sp   | sp   | ct   | ct   | sp   | sp   | ct   | ct   |
| (2)  | ct   | ct   | ct   | sp   | ct   | sp   | ct   | sp   | ct   | sp   |
| (3)  | ind  | ct   | ind  | ind  | ind  | sct  | sb   | ind  | ind  | ind  |
| (4)  | ct   | ind  | sct  | ind  | ind  | ind  | ind  | ind  | ind  | sb   |
| (5)  | sb   | ind  | sp   | ind  | ind  | ind  | ind  | ind  | ind  | ct   |
| (6)  | ind  | sb   | ind  | ind  | ind  | sp   | ind  | ind  | ind  | ind  |
| (7)  | sb   | sb   | sb   | sct  | sb   | sct  | sb   | sct  | sb   | sct  |
| (8)  | sb   | sb   | sct  | sct  | sb   | sb   | sct  | sct  | sb   | sb   |
| (9)  | cd   | ind  | sct  | sb   | ind  | ind  | ind  | ind  | sct  | sb   |
| (10) | ind  | cd   | ind  | sb   | ind  | sct  | sb   | ind  | sct  | ind  |

Here is a complete list of the 27 embedded squares occurring in the logical icosagon, where each ordered set of four vertices *a-b-c-d* is such that *a-b* are contraries, *c-d* are subcontraries, *a-d* and *b-c* are contradictories, and *c* and *d* are respective subalterns of *a* and *b*:

SQ1: (1)-(2)-(7)-(8); SQ2: (1)-(4)-(5)-(8); SQ3: (1)-(6)-(3)-(8); SQ4: (1)-(11)-(9)-(8);
SQ5: (1)-(12)-(10)-(8); SQ6: (1)-(15)-(18)-(8); SQ7: (1)-(16)-(17)-(8); SQ8: (1)-(19)-(14)-(8)
SQ9: (1)-(20)-(13)-(8); SQ10: (2)-(3)-(6)-(7); SQ11: (2)-(5)-(4)-(7); SQ12: (2)-(11)-(9)-(7)
SQ13: (2)-(12)-(10)-(7); SQ14: (2)-(13)-(20)-(7); SQ15: (2)-(15)-(18)-(7); SQ16: (2)-(17)-(16)-(7)
SQ17: (2)-(19)-(14)-(7); SQ18: (3)-(12)-(10)-(8); SQ19: (4)-(11)-(9)-(5); SQ20: (5)-(20)-(13)-(4)

SQ21: (11)-(14)-(19)-(9); SQ22: (11)-(20)-(13)-(9); SQ23: (12)-(14)-(19)-(10); SQ24: (12)-(17)-(16)-(10)

SQ25: (14)-(15)-(13)-(19); SQ26: (15)-(17)-(16)-(13); SQ27: (15)-(20)-(13)-(18).

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
