# Peer review of "A Relational Semantics for Ockham’s Modalities"

_axioms, doi:10.3390/axioms12050445_

Round 1
Reviewer 1 Report
See detailed comments on content and formulation in the attached PDF document.

Author Response
(1) [Falessi] In 214-215 you have both de dicto and de re readings. I divided them in a better way, following your suggestion. Accordingly, in 257-260 there are the inferences both in de dicto and de re readings.
(2) [Falessi] I prefer 'incompatibility' or the sake of clarity and historical accuracy. Indeed, Ockham does not call the relation between A, E and Y 'contrariety', but he uses just the term 'incompatibility'.
(3) [Schang] I agreed with the objection to using the word "bistring", and I thereby substituted it with "numbering" in our updated version;
(4) [Schang] I disagreed with the irrelevance of my three logical polygons (two dodecagons and one icosagon): I explained in a footnote that an exhaustive list of the logical relations is given in Appendix 2;
(5) [Schang] I corrected the number of normal squares in the logical icosagon: there are not 25, but 27 squares; I gave a description of these 27 items at the end of Appendix 2.
For all the other points, please see the attachment.

Reviewer 2 Report
Interesting paper, but there still are some things not clear to me, maybe because of my ignorance. It would be great if the authors could clarify these items.
First some misprints:
line 180: re-written [instead of: re-write]
line 204: how to read the symbol / in this line?
line 253: just like a disjunction implies the logical disjunction; this sentence is not clear to me.
line 318: draw [instead of: drawn]
line 401/402: A --> B = - (A ^ - B); There should be a negation sign before B
line 536: corresponds [with s]
line 604: any of the first formula X is a of the second formula Y. What do you mean?
line 671 and 675: equivalence relation [instead of: equivalent]
Next some items that are not clear to me:
lines 75-76 about the mereological distinction: could you make this more clear to me?
line 107: S is in the accusative case and P is a verb in an infinitive verb: what do you mean?
line 178: the three dotted lines with their meaning (subcontrary, contradictory and contrary) are hard to distinguish.
lines 485 - 488: I may be confused, but it seems to me that the de dicto statements should start with a quantification over possible worlds and the de re statements with a quantification over individuals.
line 586: if there is a world. By the way, I have the inclination not to accept the Barcan formula, in other words, that different worlds may contain different individuals.
By the way, the pictures are beautiful!
Author Response
(1) [Falessi] Mereological distinction. The de dicto reading takes the proposition as a whole. The de re reading divides it so that it requires to verify whether S can be said to be necessarily/possibly P or not. So, if a part of the sentence, i.e. the subject S, can be said to be, for instance, necessarily P, it does not entail that the proposition taken as a whole can be said to be necessary as well.
Take for example: “It is necessary that every truth is true”. This proposition taken as a whole is an identity, so it is necessarily true de dicto. However, a particular truth x included in the subject “every truth” is not necessarily true. So, it is false de re.
(2) [Falessi] S is in the accusative case and P is a verb in an infinitive verb: what do you mean? This is the structure of a de dicto sentence in Latin, so with modal operator introducing an infinitive sentence. The infinitive sentence has the verb as an infinite and the subject as an accusative.
(3) [Falessi] Only contraries and subcontraries have dotted lines and I do not think I can make them any clearer for graphical reasons.
(4) [Schang] I made due corrections with the typos mentioned by him/her.
Fo all the other points, please see the attachment.
